# Impact of the Anticholinergic Burden on Disease-Specific Symptoms in Parkinsonian Syndromes

**DOI:** 10.3390/brainsci14080805

**Published:** 2024-08-11

**Authors:** Romina Mahmoudi, Stephan Greten, Linda Veith Sanches, Lea Krey, Sarana Ulaganathan, Günter U. Höglinger, Johannes Heck, Florian Wegner, Martin Klietz

**Affiliations:** 1Department of Neurology, Hannover Medical School, Carl-Neuberg-Straße 1, 30625 Hannover, Germany; romina.mahmoudi@stud.mh-hannover.de (R.M.); veithsanches.linda@mh-hannover.de (L.V.S.); krey.lea@mh-hannover.de (L.K.); ulaganathan.sarana@mh-hannover.de (S.U.); guenter.hoeglinger@med.uni-muenchen.de (G.U.H.); wegner.florian@mh-hannover.de (F.W.); klietz.martin@mh-hannover.de (M.K.); 2Department of Neurology, University Hospital of Munich, Ludwig-Maximilians-Universität (LMU) Munich, 80539 Munich, Germany; 3Munich Cluster for Systems Neurology (SyNergy) Munich, 80539 Munich, Germany; 4German Center for Neurodegenerative Diseases (DZNE), 80539 Munich, Germany; 5Institute for Clinical Pharmacology, Hannover Medical School, Carl-Neuberg-Straße 1, 30625 Hannover, Germany; heck.johannes@mh-hannover.de

**Keywords:** anticholinergic burden, Parkinsonian syndromes, polypharmacy

## Abstract

**Background**: Anticholinergic adverse effects pose a relevant threat to patients, in particular elderly and cognitively impaired patients. Patients with Parkinsonian syndromes are especially at risk from anticholinergic adverse effects due to the often-required complex drug therapy. **Aims**: The aim of this study was to evaluate the potential effect of the anticholinergic burden on motor and non-motor symptoms in Parkinson’s disease and atypical Parkinsonian syndromes. **Methods:** This cross-sectional, monocentric retrospective data analysis included 151 patients with Parkinson’s disease (PD), 63 with progressive supranuclear palsy (PSP), and 36 with multiple system atrophy (MSA). The anticholinergic burden of patients’ medications was determined using two established scores: the Anticholinergic Drug Scale (ADS) and the German Anticholinergic Burden Scale (GABS). These scores were compared between the different diseases and correlated with several disease-specific scores. **Results**: Anticholinergic burden was higher in patients with PD, in particular, compared to PSP. In the PD group, anticholinergic burden showed a weak correlation with almost all analyzed clinical scores and the number of administered drugs. The UMSARS I and II showed a significant correlation with the anticholinergic burden in MSA patients. In general, the GABS-measured anticholinergic burden was significantly higher compared to the ADS-measured. **Conclusions**: The calculated anticholinergic burden affected motor and non-motor symptoms in patients with various Parkinsonian syndromes poorly. Since the GABS also contains basic anti-parkinsonian drugs, this score tended to overestimate the anticholinergic burden in patients with Parkinsonian syndromes and, therefore, seemed less appropriate for this application.

## 1. Introduction

Drugs that have an intended or accidental anticholinergic (synonym: antimuscarinic) potential can cause various adverse drug reactions (ADR), particularly cognitive and neuropsychiatric effects such as delirium [1,2,3]. Due to their altered pharmacodynamics and pre-existing comorbidities, the elderly are more susceptible to anticholinergic ADR [4,5,6]. The possible need to prescribe anticholinergic drugs for typical comorbidities in the elderly, such as depression and incontinence, completes a vicious circle [7,8]. Anticholinergic ADR not only poses an assumed threat to patient safety but can also result in real limitations in activities of daily living and increase mortality [9]. For this reason, the anticholinergic burden on elders should be minimized to ensure patient safety.

Parkinsonian syndromes, such as Parkinson’s disease (PD), progressive supranuclear palsy (PSP), and multiple system atrophy (MSA), are diseases of the elderly population [10,11,12]. In addition to motor symptoms, disease-specific non-motor symptoms, such as dementia, depression, or bladder dysfunction, and the presence of a considerable number of comorbidities complicate the required drug treatment [13,14,15]. Consequently, polypharmacy, defined as the use of five or more medications, is not uncommon [6,13,14]. Polypharmacy, in turn, has been shown to augment the anticholinergic burden and, thus, the risk of anticholinergic ADR [16,17]. So far, there are no studies that have analyzed the influence of anticholinergic burden on motor and non-motor symptoms in various Parkinsonian syndromes.

The aim of this study was to assess the anticholinergic burden in a well-defined cohort of patients with PD, PSP, and MSA and explore the potential impact on disease-specific symptoms.

## 2. Methods

### 2.1. Participants

In this monocentric retrospective study, patients were recruited in the Department of Neurology at Hannover Medical School from January 2020 to November 2023, either in neurological wards or in the specialist outpatient department for movement disorders. The local ethics committee approved the prospective data collection in accordance with the Declaration of Helsinki (no. 8666_BO_K_2019).

A total of 151 patients with PD, 63 with PSP, and 36 with MSA were analyzed. All patients gave their written, informed consent. A movement disorder specialist performed the recruitment, confirmed the clinical diagnosis, and reviewed the clinical data. The diagnosis of PD, PSP, and MSA was based on the established diagnostic criteria [18,19,20].

### 2.2. Anticholinergic Burden Measures

Two established scores, the Anticholinergic Drug Scale (ADS) and the German Anticholinergic Burden Scale (GABS), were used to determine the anticholinergic burden of the patients’ drug treatment [21,22].

The international ADS is an anticholinergic rating scale that quantifies anticholinergic burden by serum anticholinergic activity (SAA) [21,23]. Carnahan et al. verified ADS in a secondary analysis of data from patients participating in an observational study of delirium in rural long-term care facilities. In their study, the anticholinergic potency of each drug was assessed using ADS, with scores ranging from 0 (no known anticholinergic properties) to 3 (markedly anticholinergic properties) and correlated with SAA in blood samples obtained on day 14 [21].

The GABS score was, for the first time, used to assess the anticholinergic burden in patients with Parkinsonian syndromes. In contrast to the ADS, the GABS was designed based on systematic literature research on seven previously published anticholinergic burden scores. The 692 drugs originally included were reduced to the 504 drugs available in Germany. In this tool, drugs are rated in 4 levels: no (0), weak (1), moderate (2), and strong (3) anticholinergic effects [22]. 

The anticholinergic potential of chronic medication on the day of clinical evaluation was analyzed using the named scores. In addition, the potential influence of on-demand medication administered on the same day was included. Drugs with the same active ingredient were only considered once. The definition of polypharmacy was based on the number of drugs, namely the administration of five or more medications [24]. The anticholinergic exposure was categorized as follows: 0 indicated absent exposure; 1–2 indicated low exposure; and ≥3 indicated high exposure [25].

### 2.3. Clinical Assessments

The Movement Disorder Society-Sponsored Revision of the Unified Parkinson Disease Rating Scale (MDS-UPDRS), the Parkinson’s Disease Questionnaire (PDQ-8), the Geriatric Depression Scale (GDS-15), the Montreal Cognitive Assessment (MoCA) and the Scale for Evaluation of Neuropsychiatric Disorder in Parkinson’s disease (SEND-PD) were performed by a movement disorder specialist supported by a study nurse. The MDS-UPDRS is a comprehensive assessment scale for motor and non-motor symptoms in Parkinson’s patients [26]. The PDQ-8 is a questionnaire used to assess the quality of life of people with PD [27]. The GDS-15 rates depressive symptoms in older people [28]. The MoCA test is a tool to screen for cognitive impairment [29]. Lastly, the SEND-PD scale was developed to test neuropsychiatric symptoms in patients with PD [30]. Disease-specific scores such as the Unified Multiple System Atrophy Rating Scale (UMSARS I + II) and the Progressive Supranuclear Palsy Rating Scale (PSPRS) were additionally performed in MSA and PSP patients [31,32]. 

### 2.4. Analyses

IBM SPSS 29 (Armonk, NY, USA) and GraphPad PRISM 10 (San Diego, CA, USA) were used for statistical analysis. The frequencies for nominal data and the mean and standard deviation for metric data were determined and displayed. Nominal data were analyzed using the Chi-squared test. To test for normal distribution, the Shapiro–Wilk test was carried out. Since all metric data were not normally distributed, the Kruskal–Wallis test was performed for mean value comparisons, supplemented by Dunn’s multiple comparison test, and Spearman’s correlation was used to measure linear correlation. The total ADS and GABS scores, the anticholinergic burden, and the total number of drugs were correlated with the above-named clinical scores, the disease duration, and the patients’ age. 

## 3. Results

### 3.1. Patient Characteristics

Patient demographics and baseline characteristics are shown in Table 1. A total of 151 PD patients, 63 PSP patients, and 36 MSA patients were included in the study. On average, MSA patients were younger than both PD and PSP patients (PD 67.7 ± 10.96 years, PSP 69.6 ± 8.5, MSA 61.9 ± 8.9; Kruskal–Wallis test, *p* < 0.001). There was no significant difference between the cohorts in terms of sex distribution (PD 63 females (41.7%), PSP 29 (46.0%), MSA 20 (55.6%); Chi-squared test, *p* = 0.52). Compared to the PSP and MSA patients, PD patients showed significantly longer disease duration (PD 6.16 years ± 6.54 years, PSP 0.5 ± 0.9, MSA 1.1 ± 1.7; Kruskal–Wallis test, *p* < 0.001). Within the PD group, there were 76 patients with predominantly akinetic-rigid disease (50.3%), 23 patients with tremor-dominant disease (15.2%), and 51 patients with an equal distribution of motor features (33.8%). A total of 50 PSP patients (79.4%) presented with Richardson’s syndrome and 7 (11.1%) with PSP variants. The MSA cohort included 19 patients with the Parkinsonian subtype (52.8%) and 15 patients with the cerebellar subtype (41.7%). Data from 1 PD, 6 PSP, and 2 MSA patients were missing regarding the clinical phenotype. 

### 3.2. Anticholinergic Burden in Parkinsonian Syndromes

The analyses of the anticholinergic burden are displayed in Figure 1. PD patients were administered significantly more drugs than MSA patients (PD 5.8 ± 3.4 drugs per patient, PSP 4.9 ± 2.9, MSA 4.1 ± 3.2; Kruskal–Wallis test, *p* = 0.02). Overall, the PSP, MSA, and PD groups did not differ significantly in terms of ADS (PD 0.5 ± 1.1, PSP 0.4 ± 0.8, MSA 0.7 ± 1.2) and ADS burden (PD 0.1 ± 0.2, PSP 0.1 ± 0.1, MSA 0.2 ± 0.3). The GABS showed a significant difference for PSP compared to PD (PD 2.7 ± 1.7, PSP 2.0 ± 1.9, MSA 2.1 ± 1.8; Kruskal–Wallis test, *p* < 0.001). The anticholinergic exposure calculated on the basis of ADS showed no difference between the groups. In contrast, patients with PD displayed more frequent high anticholinergic exposure than PSP and MSA patients when based on GABS (Chi-squared test, *p* = 0.002). 

### 3.3. Influence of Anticholinergic Burden on Clinical Symptoms in PD

The ADS and GABS, the respective burden, and the number of administered drugs were correlated with commonly used scores, duration of disease, and age, as shown in Table 2. The number of drugs showed a significant correlation with age, disease duration, and scores reflecting the patient’s cognition, non-motor and motor symptoms, as well as quality of life. 

ADS significantly correlated with the disease duration, and ADS additionally correlated with the number of drugs. The GABS showed a significant positive correlation with almost all clinical scores except MoCA and GDS-15. The five most commonly used anticholinergic drugs in PD patients evaluated with ADS were amantadine, clonazepam, sertraline, prednisolone, and clozapine. For GABS, these were levodopa, domperidone, pramipexole, rotigotine, and entacapone.

### 3.4. Influence of Anticholinergic Burden on Clinical Symptoms in Atypical Parkinsonian Syndromes

The correlation results for PSP and MSA patients are shown in Table 3 and Table 4. In PSP patients, the ADS and GABS showed a strong positive correlation with the number of drugs. The five most common anticholinergic drugs in PSP patients evaluated with ADS were amantadine, lorazepam, clonazepam, furosemide, and olanzapine, and with GABS, levodopa, amantadine, domperidone, metoprolol, and mirtazapine.

MSA patients offered a negative correlation between numerous drugs and the GDS-15. There was a positive correlation between ADS and UMSARS I and II. As expected, ADS and GABS correlated with the number of administered drugs. Moreover, the GABS showed a positive correlation with disease duration. For the MSA group, the five most common medications assessed with ADS were amantadine, amitriptyline, chlorthalidone, clonazepam, and darifenacin, and with the GABS, levodopa, amantadine, domperidone, venlafaxine, and metformin. 

## 4. Discussion

In this study, the association of the anticholinergic burden with disease-specific symptoms was investigated in a large and clinically well-defined cohort of patients with PD and atypical Parkinsonian syndromes. Patients with PD showed the highest anticholinergic burden, which was presumably due to the significantly higher number of drugs administered and the composition of the scores used, particularly the GABS. Furthermore, a higher anticholinergic burden was associated with a poorer clinical condition in terms of motor and non-motor symptoms in PD. Interestingly, the cognitive function measured by the MoCA was unaffected by the anticholinergic burden. The most common anticholinergic drugs recorded were amantadine, clonazepam, and dopaminergic agents (e.g., levodopa).

To date, several studies have been published concerning anticholinergic burden in PD patients, but none for PSP and MSA patients [33,34,35]. 

Nawaz et al. calculated the anticholinergic burden using the Anticholinergic Cognitive Burden Scale (ACB), Anticholinergic Risk Scale (ARS), Drug Burden Index–Anticholinergic Component (DBI-Ach), and ADS in outpatients with PD [33]. The outpatient cohort showed an overall higher prevalence of PD patients receiving anticholinergic medication (46.3% in Nawaz et al. vs. 26.5% in our cohort) and, in addition, a distribution in favor of a higher anticholinergic burden (11.6% with high burden in Nawaz et al. vs. 6.6% in our cohort). The higher age of the 670 included patients and/or the administration of more drugs per patient could be responsible for this [33]. Both factors, but especially the latter, are closely linked to a higher anticholinergic burden [16,17,34].

As previously described, anticholinergic side effects, particularly of a cognitive and neuropsychiatric nature, pose a potential threat to patients [17,36,37]. A correlation between the assessed cognitive performance using the MoCA and the anticholinergic burden was not found in the analysis carried out. This result was similar to Sumbul-Sekerci et al., although it should be pointed out that that group chose the reverse analysis approach by comparing the anticholinergic burden between PD patients with and without mild cognitive impairment (MCI) [34]. The study by Rajan et al. revealed an inverse relationship [35]. Here, a higher anticholinergic burden had a significant association with cognitive impairment and freezing of gait (FOG). Compared to Sumbul-Sekerci et al., who diagnosed MCI based on the MDS Level 2 comprehensive assessment, and the study conducted here, which used the MoCA as an established cognitive test, the findings of Rajan et al. were based on single, less comprehensive items of the MDS-UPDRS, specifically item 1.1 for cognitive impairment and 2.13 and 3.11 for FOG [34,35,38]. Pfistermeister et al. identified a strong negative interrelation between a high anticholinergic burden determined by ACB and cognitive function, as well as the diagnosis of dementia, in a cohort of 59,007 geriatric patients. Crispo et al. affirmed an interaction between anticholinergic burden and cognitive or neuropsychiatric outcome by detecting increased rates of delirium in PD patients with a high anticholinergic load assessed by ARS [36]. Furthermore, this particular group of patients had a higher number of emergency admissions and readmissions after discharge. Suggested explanations for the missing correlation between anticholinergic and cognitive function in our cohort could be the rather small number of patients, the lack of consideration of the usage duration and dosage of the drug, and the current lack of follow-up evaluations.

Besides the pharmacodynamically derivable peripheral and central nervous adverse effects, potential anticholinergic drugs appear to cause additional adverse effects that defy obvious explanations, such as increasing the risk of falls [17,39,40]. This non-negligible issue poses a special threat to patients with Parkinsonian syndromes, in particular atypical Parkinsonian syndromes such as PSP and MSA. Postural instability is a classic symptom in the natural course of PD [41,42]. In contrast, falls occur earlier and more frequently in patients with atypical Parkinsonian syndrome, so these have become part of the diagnostic criteria for PSP [19,43,44,45]. However, such assumptions are currently theoretical, as there are no prospective studies in Parkinsonian syndromes on the influence of anticholinergic burden on the actual occurrence of anticholinergic side effects other than cognitive ones.

To our knowledge, this is the first study to use the GABS to assess anticholinergic burden in patients with Parkinsonian syndromes. The drugs classified in the GABS were compiled based on data from seven established scores, including the ADS and the ARS, and adapted for the German drug market [22]. The quality of the GABS has already been confirmed in a literature-based comparison with various scores [46]. Although the rigor of the development was considered rather questionable, the GABS performed better than most in terms of editorial independence, stakeholder involvement, and clarity of presentation. The results of this solely statistical view were validated in a prospective study of a large cohort of older patients in comparison to ADS [47]. As the ARS was the only previously established anticholinergic classification incorporating dopaminergic drugs such as levodopa, these are also integrated into the GABS [22,48]. This structural difference probably explains why the scores for GABS in our study were consistently higher than those for ADS, which, apart from amantadine, does not contain any medication approved for Parkinsonian syndromes. Previous studies investigating the reliability between the different anticholinergic burden scales showed a comparatively low to medium concordance between ARS and ADS [25,49]. Ultimately, the question arises as to whether the GABS is a suitable tool for assessing anticholinergic burden in patients with Parkinsonian syndromes. If the classic anti-parkinsonian drugs (Anatomical Therapeutic Chemical Classification System (ATC) N04), e.g., levodopa and dopamine agonists, are recognized as having the anticholinergic potency postulated by Rudolph et al. in the ARS, the GABS is certainly more sensitive than other scales for identifying potential anticholinergic burden, with the risk of overinterpretation [48]. However, since the anticholinergic effect of anti-parkinsonian drugs, with the exception of amantadine, is known to be insignificant, other scores appear to more accurately depict the real anticholinergic load [6].

## 5. Limitations

A limitation of this study is the rather low number of patients due to the monocentric design. In addition, although the medication on the day of the assessments was used for the correlation with the established disease-specific scores and cognitive performance, neither the exposure time nor the dosage of the anticholinergic medication were taken into account as potential factors influencing the anticholinergic load [50,51]. Lastly, a potential limitation of the study could be the focus on just two scales to measure the anticholinergic burden, although several other scales are available. From another perspective, however, this may also be a benefit, as it improves clarity and transparency. Moreover, both scales were selected for the following reasons: the GABS represents an up-to-date scale tailored to the German drug market for the first time and integrates a number of scales that have already been established for many years, e.g., ARS and ACB. The ADS, on the other hand, provides a rather objective scale for comparison, as it correlates the anticholinergic burden to actual serum concentrations.

## 6. Conclusions

Given the demographic characteristics and the inevitably complex drug therapy frequently requiring polypharmacy, patients with Parkinsonian syndromes are particularly susceptible to a considerable anticholinergic burden. As a substantial anticholinergic burden can deteriorate disease-specific symptoms, a critical evaluation of potential anticholinergic drugs should be undertaken as part of holistic treatment.

Although the GABS detects very sensitively the presence of anticholinergic exposure, this classification rather exaggerates the severity of anticholinergic stress in Parkinsonian syndromes, as it also contains basic anti-parkinsonian drugs.

Further prospective studies on the real-life impact of anticholinergic burden on the occurrence of anticholinergic side effects and outcomes in patients with Parkinsonian syndromes are urgently needed.

## Figures and Tables

**Figure 1 brainsci-14-00805-f001:**
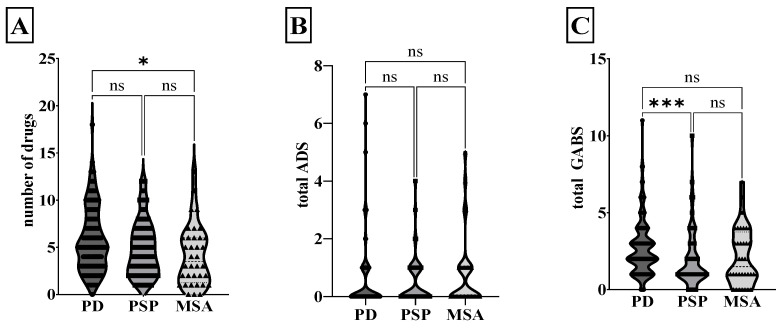
Anticholinergic burden in Parkinsonian syndromes. The violin plot shows the median and the quartiles of the number of drugs (**A**), total ADS (**B**), and total GABS (**C**) for PD, PSP, and MSA. * *p* < 0.05, *** *p* < 0.001, ns non-significant, Kruskal–Wallis test. GABS, German Anticholinergic Burden Scale; ADS, Anticholinergic Drug Scale; MSA, multiple system atrophy; PD, Parkinson’s disease; PSP, progressive supranuclear palsy.

**Table 1 brainsci-14-00805-t001:** Patients’ Characteristics.

	PD (n = 151)	PSP (n = 63)	MSA (n = 36)
Age in years, mean ± SD	67 ± 10.96 **	69.4 ± 8.5 ***	61.9 ± 8.8
Sex, female, n (%)	63 (41.7)	29 (46.0)	20 (55.6)
Disease duration in years, mean ± SD	6.16 ± 6.54 ***	0.5 ± 0.9	1.1 ± 1.7
Clinical phenotype, n (%)			
PD Tremor-dominant	23 (15.2)		
PD Akinetic-rigid	76 (50.3)		
PD Equivalent	51 (33.8)		
PSP-Richardson Syndrome		50 (79.4)	
PSP non-Richardson Syndrome		7 (11.1)	
MSA-P			19 (52.8)
MSA-C			15 (41.7)
Missing	1 (0.7)	6 (9.5)	2 (5.6)
Number of drugs, mean ± SD	5.8 ± 3.4	4.9 ± 2.9	4.1 ± 3.2 *
Patients with polypharmacy, n (%)	91 (60.3)	31 (49.2)	16 (44.4)
Anticholinergic exposure, ADS, n (%)			
absent	111 (73.5)	42 (66.7)	22 (61.1)
low	30 (19.9)	19 (30.1)	10 (28.8)
high	10 (6.6)	2 (3.2)	4 (11.1)
Anticholinergic exposure, GABS, n (%)			
absent	5 (3.3)	8 (12.7)	7 (19.4)
low	77 (51.0)	39 (61.9)	14 (38.9)
high	69 (45.7)	16 (25.4)	15 (41.7)

* *p* < 0.05, ** *p* < 0.01, *** *p* < 0.001, Kruskal–Wallis test. GABS, German Anticholinergic Burden Scale; ADS, Anticholinergic Drug Scale; MSA, multiple system atrophy; MSA-C, cerebellar variant of multiple system atrophy; MSA-P, Parkinsonian variant of multiple system atrophy; PD, Parkinson’s disease; PSP, progressive supranuclear palsy; SD, standard deviation.

**Table 2 brainsci-14-00805-t002:** Impact of anticholinergic burden on established scores in PD.

	Age	Disease Duration	MoCA	UPDRS I	UPDRS II	UPDRS III	UPDRS Total	SEND-Psycho	SEND-PD Total	GDS-15	PDQ-8	Number of Drugs
Number of drugs	0.431 **	0.167 **	−0.302 **	0.341 **	0.340 **	0.396 **	0.438 **	0.167	0.130	0.160	0.218 *	
ADS	0.011	0.172 *	−0.108	0.169	0.169	0.038	0.134	0.021	0.111	0.033	0.046	0.374 **
GABS	0.133	0.296 **	−0.153	0.357 **	0.344 **	0.248 **	0.374 **	0.267 **	0.262 **	0.156	0.252 **	0.517 **

Rho: <0.3 no or weak correlation, 0.3–0.5 moderate correlation, >0.5 strong correlation; * *p* ≤ 0.05, ** *p* ≤ 0.01; Spearman correlation.

**Table 3 brainsci-14-00805-t003:** Impact of anticholinergic burden on established scores in PSP.

	Age	Disease Duration	MoCA	PSPRS	SEND-Psycho	SEND-PD Total	GDS-15	PDQ-8	Number of Drugs
Number of drugs	−0.022	0.064	0.032	−0.066	−0.082	−0.2	−0.024	0.147	
ADS	−0.058	0.163	−0.037	−0.134	−0.010	0.139	−0.091	0.163	0.435 **
GABS	−0.011	0.057	−0.053	−0.024	−0.065	0.125	−0.154	0.24	0.389 **

Rho: <0.3 no or weak correlation, 0.3–0.5 moderate correlation, >0.5 strong correlation; ** *p* ≤ 0.01; Spearman correlation.

**Table 4 brainsci-14-00805-t004:** Impact of anticholinergic burden on established scores in MSA.

	Age	Disease Duration	MoCA	UMSARS I + II	SEND-Psycho	SEND-PD Total	GDS-15	PDQ-8	Number of Drugs
Number of drugs	0.231	0.151	−0.331	0.327	0.182	−0.012	−0.486 **	−0.009	
ADS	0.087	0.079	−0.197	0.386 *	0.074	0.036	0.062	−0.78	0.442 **
GABS	0.106	0.349 *	−0.182	0.305	0.014	0.055	−0.228	0.098	0.704 **

Rho: <0.3 no or weak correlation, 0.3–0.5 moderate correlation, >0.5 strong correlation; * *p* ≤ 0.05, ** *p* ≤ 0.01; Spearman correlation.

## Data Availability

The data supporting the findings of this study are available from the corresponding author, Dr. Stephan Greten, upon reasonable request.

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
