# Peer review of "Impact of the Anticholinergic Burden on Disease-Specific Symptoms in Parkinsonian Syndromes"

_brainsci, 2024, doi:10.3390/brainsci14080805_

Round 1
Reviewer 1 Report
Comments and Suggestions for Authors
Mahmoudi et al, have documented the clinical findings and analytical correlations in this study that shed light on anticholinergic ADR during parkinsonism treatments. They clearly state the methods, purpose, and outcomes of the study. In the discussion section, they successfully compare their results with other studies in the field, which also serve as a good literature summary.
Some points of clarification or things to consider:-
In Table 1, consider if it would be more fitting to change the levels of GABS in the original 4-stage system (no, low, moderate, high). If not, explain how the numbers were grouped to this three-tier system.
To understand the spread of the data in figure-1, It might be visually more informative to convert the bar graphs to violin plots by showing the individual values.
In the section where five drugs are listed, please explain why the list is completely (or 80% for non-PD cases, except amantadine) complementary without any overlap. Are any of the top ADS drugs not listed under GABS?
The table 2-4 legends (especially 3 and 4) are highly redundant. Consider either consolidating them or omitting them, especially since they are also explained in the text earlier. In place, it might be visually informative to add the actual correlation plots for at least ADS , GABS vs (i) number of drugs (possibly in the same graph, two colored lines) for all three diseases and (ii) One of the disease scores.
Line-216: "As differing scales were used, the only meaningful comparison was that concerning ADS" - Please explain or expand the logic.
Author Response
Reviewer 1:
Mahmoudi et al, have documented the clinical findings and analytical correlations in this study that shed light on anticholinergic ADR during Parkinsonism treatments. They clearly state the methods, purpose, and outcomes of the study. In the discussion section, they successfully compare their results with other studies in the field, which also serve as a good literature summary.
Some points of clarification or things to consider:
Comment 1: In Table 1, consider if it would be more fitting to change the levels of GABS in the original 4-stage system (no, low, moderate, high). If not, explain how the numbers were grouped to this three-tier system.
Answer 1: Thank you for this comment. The classification of the anticholinergic burden using GABS is drug-related, so that each drug is categorized between 0 and 3 according to the strength of the anticholinergic effect. In contrast, the anticholinergic exposure shown in Table 1 is patient-related and is based on the work of Pont and colleagues (reference number 25). The sum of the individual anticholinergic load is calculated and divided into the categories absent, low and high (see section “methods” lines 93 to 95). Since patient-related characteristics of the populations are listed in Table 1, it is thus only possible to show the patient-related anticholinergic exposure.
Comment 2: understand the spread of the data in figure-1, It might be visually more informative to convert the bar graphs to violin plots by showing the individual values.
Answer 2: Many thanks for this very constructive comment. Figure 1 was changed to a violin plot.
Comment 3: In the section where five drugs are listed, please explain why the list is completely (or 80% for non-PD cases, except amantadine) complementary without any overlap. Are any of the top ADS drugs not listed under GABS?
Answer 3: This is a very interesting question. In fact, the situation is reversed, because the differences are based on the composition of the GABS. Since the GABS, unlike the ADS, contains standard Parkinson's medications and these were of course used regularly in our cohort, this class of drugs is represented disproportionately frequently.
Comment 4: The table 2-4 legends (especially 3 and 4) are highly redundant. Consider either consolidating them or omitting them, especially since they are also explained in the text earlier. In place, it might be visually informative to add the actual correlation plots for at least ADS , GABS vs (i) number of drugs (possibly in the same graph, two colored lines) for all three diseases and (ii) One of the disease scores.
Answer 4: Thank you very much for the helpful hint. This improves the clarity considerably. The legends of the tables have been adjusted. The suggestion of a correlation plot also seems very useful. In order to keep the structure of the paper consistent and since it would not contain any new information, such a figure was not included.
Comment 5: Line-216: "As differing scales were used, the only meaningful comparison was that concerning ADS" - Please explain or expand the logic.
Answer 5: Again, thank you for the helpful advice. The sentence was removed from the manuscript as it does not offer any relevant additional content and even leads to misunderstanding.
The aim of this sentence was to point out that many scores were used in the aforementioned study to assess anticholinergic stress that were not used in our study. Only the ADS was used in both studies and was therefore comparable. However, as the scores used in the study have already been mentioned, this sentence does not provide any further information.
Reviewer 2 Report
Comments and Suggestions for Authors
The work concerns the important topic of the use of anticholinergic drugs in the population of patients affected by parkinsonism. Due to their age, comorbidities and the fact that drugs used in Parkinson's disease have anticholinergic potential, these patients are particularly vulnerable to side effects of these drugs.
What is particularly noteworthy is that the authors gathered a large group of patients with atypical parkinsonisms. For which there is very little data on the possible effects of cholinergic burdon. The work brings new valuable information to the discussion about cholinergic burden in parkinsonisms. Further research focused on problems specific to this group (such as falls or cognitive disorders with a broader neuropsychiatric assessment) is necessary to fully explain the significance of this phenomenon.
In PD group there are several patient characteristics, e.g., age, that are significantly different so p values of factors of particular interest, e.g., anticholinergic burden, should be adjusted for these. Was cognitive assessment based solely on MoCA? Were other factors taken into account? The choice of scales assessing cholinergic burden should be discussed (its strength and limitations), other scales such as the Anticholinergic Cognitive Burden Scale (ACB) or the CRIDECO Anticholinergic Burden Scale (CALS) could also be valuable in this case.
Author Response
Reviewer 2:
The work concerns the important topic of the use of anticholinergic drugs in the population of patients affected by Parkinsonism. Due to their age, comorbidities and the fact that drugs used in Parkinson's disease have anticholinergic potential, these patients are particularly vulnerable to side effects of these drugs.
What is particularly noteworthy is that the authors gathered a large group of patients with atypical Parkinsonism. For which there is very little data on the possible effects of cholinergic burden. The work brings new valuable information to the discussion about cholinergic burden in Parkinsonism. Further research focused on problems specific to this group (such as falls or cognitive disorders with a broader neuropsychiatric assessment) is necessary to fully explain the significance of this phenomenon.
Comment 1: In PD group there are several patient characteristics, e.g., age, that are significantly different so p values of factors of particular interest, e.g., anticholinergic burden, should be adjusted for these.
Answer 1: We very much welcome your comments. Unfortunately, we had trouble understanding your concerns. The cohort of PD and PSP patients differed from the MSA patients in age; in addition, the duration of the disease in PD patients differed from that of the other diseases. The only significant difference in anticholinergic burden, which was determined using GABS, was between the PD and PSP patients. There was no age difference here. All mean comparisons (Kruskal Wallis test) were corrected for multiple testing using the Dunn correction. A section was added to the methods section.
Are these explanations sufficient for you? I would be happy to make further changes if you specify your comment.
Comment 2: Was cognitive assessment based solely on MoCA? Were other factors taken into account?
Answer 2: Thank you for the question. Only the MoCA was used to perform a general assessment of the global cognitive function in our patient cohort. Since the MoCA is a long-established clinical tool for the initial assessment of cognitive function in patients with various Parkinson's syndromes, we decided to use it.
I hope that i have understood your question correctly: other factors relating to cognitive functions were not included in order to establish comparability between the different Parkinsonian syndromes.
In addition to cognitive performance, however, the influence of anticholinergic burden on other clinical symptoms, such as neuropsychiatric symptoms measured by SEND-PD and GDS-15, was investigated.
Comment 3: The choice of scales assessing cholinergic burden should be discussed (its strength and limitations), other scales such as the Anticholinergic Cognitive Burden Scale (ACB) or the CRIDECO Anticholinergic Burden Scale (CALS) could also be valuable in this case.
Answer 3: Thank you again for this interesting inquiry. The detailed decision-making process was not included in the manuscript for the sake of clarity.
Since our cohort was examined in Germany, we decided to use the GABS, as it offers a new type of scale tailored to the German drug market and based on extensive literature research, e.g. the ACB. For comparison, we chose a largely objective scale, namely the ADS, which was developed on the basis of laboratory values ​​and without expert opinion. The limitation here is certainly that the ADS has not been updated for some time and also contains fewer drugs.
I welcome the suggestion of using the ACB or the CALS, as both are based on a combination of laboratory values, literature research and expert opinion. However, the CALS and the GABS included the ACB, so a further comparison or use of the ACB did not seem to provide any additional information.
The CALS, on the other hand, is even more up-to-date and is based on the ACB and GABS, among others. Accordingly, it could provide added value in terms of information, particularly since there is an interesting current study that examined the risk of dementia in PD patients (PMID 37357543). However, there is a large overlap between the studies included in the development of the GABS and the CALS (e.g. ADS, ARS, ACB). In addition, the CALS is based on studies from Korea and New Zealand. Accordingly, in our specific case, the use of the GABS seems to make the most sense due to the location of the study and the fact that it is being used for the first time.
The lines 278 - 286 have been added to mention this limitation and to briefly state the reasons for using the scales. I hope that this point has been sufficiently addressed.
Reviewer 3 Report
Comments and Suggestions for Authors
This interesting article refers to the risk of dementia in aging patients with Parkinsonian syndromes who are treated with anti-cholinergic medications, which is a highly relevant issue considering that any aged patient is subjected to cognitive risks when treated with anti- cholinergic therapeutics. The selection of patients with Parkinsonian syndromes makes further sense given the known imbalance between acetylcholine and dopamine levels in these particular group of studied patients. This being said, the population size is somewhat too small to judge the outcome, which should perhaps be highlighted more as a putative limitation.
Author Response
Reviewer 3:
This interesting article refers to the risk of dementia in aging patients with Parkinsonian syndromes who are treated with anti-cholinergic medications, which is a highly relevant issue considering that any aged patient is subjected to cognitive risks when treated with anti- cholinergic therapeutics. The selection of patients with Parkinsonian syndromes makes further sense given the known imbalance between acetylcholine and dopamine levels in these particular group of studied patients.
Comment 1: This being said, the population size is somewhat too small to judge the outcome, which should perhaps be highlighted more as a putative limitation.
Answer 1: In the "Limitations" section, lines 284 to 285 of the manuscript already contain the following statement on the size of the population: “A limitation of this study is the rather low number of patients due to the monocentric design.” The "Conclusions" section also emphasizes the need for a prospective multicenter study to assess the actual effects of anticholinergic stress on the well-being of Parkinson's patients in line 298 to 300: “Further prospective studies on the real-life impact of anticholinergic burden for the occurrence of anticholinergic side effects and outcome in patients with Parkinsonian syndromes are urgently needed. Are these references to the small population and the resulting limited rigor of the results sufficient?